# A Secure Blockchain Framework for Storing Historical Text: A Case Study of the Holy Hadith

Khaled M. Awad [1,*], Mustafa ElNainay [1,2], Mohammad Abdeen [3], Marwan Torki [1], Omar Saif [4] and Emad Nabil [3,5]

1. Department of Computer and Systems Engineering, Alexandria University, Alexandria 21526, Egypt; melnainay@aiu.edu.eg (M.E.); mtorki@alexu.edu.eg (M.T.)
2. Faculty of Computer Science and Engineering, AlAlamein International University, Matrouh 51718, Egypt
3. Faculty of Computer and Information Systems, Islamic University of Madinah, Madinah 42351, Saudi Arabia; mabdeen@iu.edu.sa (M.A.); e.nabil@fci-cu.edu.eg (E.N.)
4. Faculty of Hadith and Islamic Studies, Islamic University of Madinah, Madinah 42351, Saudi Arabia; osaif@iu.edu.sa
5. Faculty of Computers and Artificial Intelligence, Cairo University, Giza 12613, Egypt
* Correspondence: khaledbnmohamed@hotmail.com

**Abstract:** Historical texts are one of the main pillars for understanding current civilization and are used to reference different aspects. Hadiths are an example of one of the historical texts that should be securely preserved. Due to the expansion of the online resources, fabrications and alterations of fake Hadiths are easily feasible. Therefore, it has become more challenging to authenticate the online available Hadith contents and much harder to keep these authenticated results secure and unmanipulated. In this research, we are using the capabilities of the distributed blockchain technology to securely archive the Hadith and its level of authenticity in a blockchain. We selected a permissioned blockchain customized model in which the main entities approving the level of authenticity of the Hadith are well-established and specialized institutions in the main Islamic countries that can apply their own Hadith validation model. The proposed solution guarantees its integrity using the crowd wisdom represented in the selected nodes in the blockchain, which uses voting algorithms to decide the insertion of any new Hadiths into the database. This technique secures data integrity at any given time. If any organization's credentials are compromised and used to update the data maliciously, 50% + 1 approval from the whole network nodes will be required. In case of any malicious or misguided information during the state of reaching consensus, the system will self-heal using practical Byzantine Fault Tolerance (pBFT). We evaluated the proposed framework's read/write performance and found it adequate for the operational requirements.

**Keywords:** holy hadith; permissioned blockchain; byzantine fault tolerance; hyperledger fabric; data integrity; secure blockchain; practical byzantine fault tolerance; islamic hadith

## 1. Introduction

Preserving historical text is crucial as many documents are valuable and link back to their roots. Some are used as a resource for regulations for religions such as Islam.

Islam is the second-largest religion globally, and its followers use the Quran and Hadith as the primary sources of ethical guidelines and principles. The daily activities, sayings, and narrations of the Prophet Muhammad (PBUH) are called Hadith or Sunnah. Hadiths are the optimal practical examples that Muslims accomplish in their daily activities. The Quran is the complete message from God (Allah) in the form of verses to humankind, and Hadiths are the practical demonstration and descriptions of Allah's message through the Prophet Muhammad. For example, in the Quran, Allah has ordered humankind to worship him (e.g., prayer—Salah), whereas Prophet Muhammad practically demonstrates and teaches the preparation and offering of prayers (i.e., steps of ablutions and prayer).

Furthermore, the Quran mentions the importance of the Hadith through different verses (e.g., Al-Ahzaab 33 (21) and An-Noor 24 (54)).

A blockchain [1] is a decentralized public ledger used to store data using a transaction-based methodology using peer-to-peer distributed storage to maintain the immutability of the data. This introduces countless possibilities for different applications [2], from cryptocurrency to automated payments without third party intervention [3], to digital content source tracking [4], and to access control and internet of things (IoT) device security protection [5,6], and to privacy preservation [6,7].

Blockchain technology is eliminating the need for a middleman to authenticate the data. Data authentication and verification processes are encouraged by a reward. This reward is based on blockchain technology but can mostly be mapped to an economical value. Any peer who wants to participate in these processes is called a miner and uses a verification algorithm for the transactions using an algorithm called "proof-of-work" [8]. The reward concept results in many users' involvement with the fundamental requirement of having a local copy of the distributed ledger. Multiple consensus protocols can be used to update the data on the blockchain's spread copies and synchronize these versions between different blockchain peers. Multiple grouped transactions are called "blocks" that define the "blockchain" name. A block consists of transactions and other block information and metadata needed for verification and data integrity, including a reference to the previous block in the blockchain. Any modification to any block will be easily discovered as it is theoretically and computationally infeasible to manipulate the data on all the existing copies of the blockchain without being discovered. A blockchain, consisting of a chain of blocks, is immutable, where the different blocks are linked to each other employing cryptography.

A consortium blockchain is an organizational blockchain technology structure with multiple organizations that govern the network instead of a single one. A single organization controls a private network, which acts as a centralized entity. In the consortium blockchain, multiple organizations can manage the decision-making in the network. This significantly decreases the possibility of illegal alteration of the data, such as adding fake/invalid Hadiths, because other organizations monitor these actions. A hybrid network that merges the possibilities of both public and private networks is called a permissioned blockchain. The use of consortium blockchain elevates collaboration between the involved organizations, the universities in our case, where all of them can share the same authority on the blockchain.

Storing Hadiths can be performed using multiple ways and with regular data storage. A base solution can be a single database. In addition to adding layers of security with the least privileges to perform actions on the data. Another popular approach is to add multiple authorization levels on both the application and the database hosting server level. The solution we are introducing is a different and new approach using the advantages of the blockchain technology mentioned above to keep data integrity across multiple nodes and be easily scalable by adding new nodes representing new data validators.

The critical feature for using blockchain in our solution is data integrity, which refers to the reliability and trustworthiness of data. It involves the maintenance of and assurance of the accuracy and consistency of data over its entire life cycle. All the efforts for analyzing Hadiths should be saved securely and maintained by the trusted institute that made these analyses.

It is vital to keep up with the rapid changes in blockchain and the new architectures that frequently appear. Using the maximum potential of the blockchain needs a broad search for the different applications suggested by the community and how to construct other solutions based on some tweaks to the already open-sourced blockchain framework presented architecture, which we will discuss more in the following sections. The rest of this paper is organized as follows: The literature background is in Section 2; the proposed blockchain-based solution is described in Section 3, and its implementation details are given in Section 4. The performance of the proposed solution is evaluated in Section 5, and the paper is concluded in Section 6.

## 2. Literature Background

Blockchain technology has started to emerge with multiple applications, such as, for example, in finance. Conventionally, an intermediary such as a bank verifies and processes the financial transactions. Being dependent on a centralized system such as this puts immense work in the hands of intermediaries. Meanwhile, the transactions might have errors as multiple uncoordinated parties need to manage the record. Thus, the entire process is cost-ineffective and time-consuming. A need for a layer of transparency was added by introducing nodes in a blockchain where each node has a copy of the updated blockchain. Blocks are chronologically arranged; therefore, when a block including verified transactions is added to the blockchain, the entire blockchain is immutable. Therefore, attackers are not able to manipulate the transactions once they are registered in the system. Moreover, it is heavily used in cryptocurrency, which holds a market cap in the billions of dollars. Specifically, Bitcoin, proposed by a person/multiple known as Satoshi Nakamoto, allows sending/receiving money securely between entities, eliminating the need for a trusted third party such as banks. Based on the concept of being cryptographically impossible to reveal a private key from its public key [9], this asymmetric encryption keeps the authenticity of the users from being impersonated by attacks. To start a transaction, the Bitcoin client application specifies the number of Bitcoins you want to send encrypted with a combination of the recipient's public key and the sender's private key. Then the transaction is sent to the distributed Bitcoin network to be verified by other peers to make sure that a valid owner sent the money by exploiting mathematical relationships between its public and private keys. If verification is successful, the transaction log is stored in every Bitcoin user's copy of the blockchain to keep track of Bitcoin credit; therefore, it is practically impossible to use another person's Bitcoins [1].

In a blockchain, the longest chain of verified records of transactions is the only chain accepted, which is powered by the blockchain community to ensure its correctness. This means that data corruption implies corrupting hundreds of copies existing on different users, distributed across different locations. A short-chain means it is backed by relatively fewer users, which means less trust; therefore, it is rejected in the blockchain and marked as a stale fork of the chain to keep its records.

In Islamic studies and finance, blockchain has been put on the table with different applications. Where Islamic finance is based on the shariah (regulations of Islam), blockchain has the potential to be enforced with Shariah compliment's smart contracts [10]. Moreover, a paper proposed a solution called "Saadiqin" [11], an industrial solution for contract-based Islamic core banking that complies with the necessary pillars and conditions related to Muamalah Financial Contracts. The introduction of blockchain creates new possibilities for financial services to harness the benefits of blockchain. This paper [11] discusses the current problems with financial services, particularly Islamic finance, the benefits of blockchain implementation in financial services, and the possibility of Saadiqin integration with blockchain. The overall structure of integration and initial progress is presented.

Blockchain can also be used in fields such as halal food supply chain management [12], including procurement, distribution, handling, processing materials, stocking, and until the final delivery to the end customer. Regulators in Islamic majority countries can monitor a blockchain system, which guarantees that the food/drinks are halal and toyyib [12], and using permissioned blockchain with different channels, the halal supply chain can be confidential, secure, and immutable, while giving the end-user a completely open and transparent view on the whole chain [13].

In [14], the available online contents of Hadith are presented from multiple sources; many of them are not verified by any kind of authority/trusted organization, so this results in the presence of many fake/wrongly classified Hadiths. A new need arose to store the results of the consumed efforts of the researchers working on the verification and authentication of the Hadiths. After providing their analysis, the authors suggested storing the resulting Hadith in a secure database such as a blockchain to protect it from any type of tampering by the attackers.

We propose a solution that preserves Hadiths from any kind of manipulation by storing the Hadiths in blocks in a blockchain managed by the main Islamic institutes that are responsible for preserving the Islamic heritage, using the authority and credibility of these institutes to build a hybrid blockchain system where it needs less computational power for these institutes to insert Hadiths into the blockchain proposed.

We used the open-source Hyperledger Fabric as the base blockchain framework for our solution to benefit from the permissioned blockchain possibilities. In addition to the ability to customize the endorsement policy based on our use case and the development community's maturity.

Table 1 provides a high-level comparison of the current traditional centralized data archives for the Holy Hadith against the proposed solution.

**Table 1.** Proposed solution features against traditional archives.

|  | Hadiths' Current Data Archives | Proposed Solution |
| --- | --- | --- |
| Reliability of the digital Hadith database | Based on the archival owner entity | Collaborative efforts of multiple entities |
| Data verification model | Single verification methodology | Multiple methodologies |
| Client Scalability | Highly scalable | Highly scalable |
| Entitles Scalability | Not available | Fairly scalable |
| Correctness proofs | Not available | Available |
| Read/Write performance | Fast | Fair |
| Data Replicas | Secondary replicas | Distributed ledgers |
| Consensus finality | Not available | Data can be permanent and immutable |

### 3. Proposed Blockchain-Based Solution

In our proposed solution, the system uses the prefabricated Hyperledger Fabric template to create new networks based on Hyperledger Fabric v1.4. Starting with three organizations and one channel, as shown in Figure 1, these can be extended as the network needs.

Every organization includes A single peer node representing the organization in the blockchain, a self-managed root certificate authority that binds the nodes to the organization, and a membership service provider responsible for all cryptographic protocols and user authentication. Each organization server contains configuration defining the consensus algorithm used, the organization's authorized members, an event management/subscription system to track the updates of the blockchain events, and a custom smart contract implementation used for transaction insertion. Finally, each organization has its own synced version of the ledger (database).

The solution includes a client-facing REST server that initially authenticates the users using a basic authentication system. The REST server provides REST APIs that can be consumed globally using web, mobile, or even CLI user interfaces.

Each organization will apply its own Hadith verification model, for example, using the power of artificial intelligence and machine learning to build an Arabic natural language processing model based on the online Hadiths' different datasets. This model can define the degree of correctness of the Hadith and then consume the blockchain backend server endpoints using the organization client to add the Hadith to the network. A recent study [15], identifies the ambiguous narrator and their reliability level. The study presents a new dataset that contains narration chains with their identified narrators, which can be integrated into the proposed blockchain.

The network is built with the official Hyperledger Fabric images for peers, orderers, and chaincodes where all peers have multiple policies (access) to different actions that allow writing and reading from/to the blockchain. We have a consortium that contains all non-orderers organizations and wallets for all peers in the network.

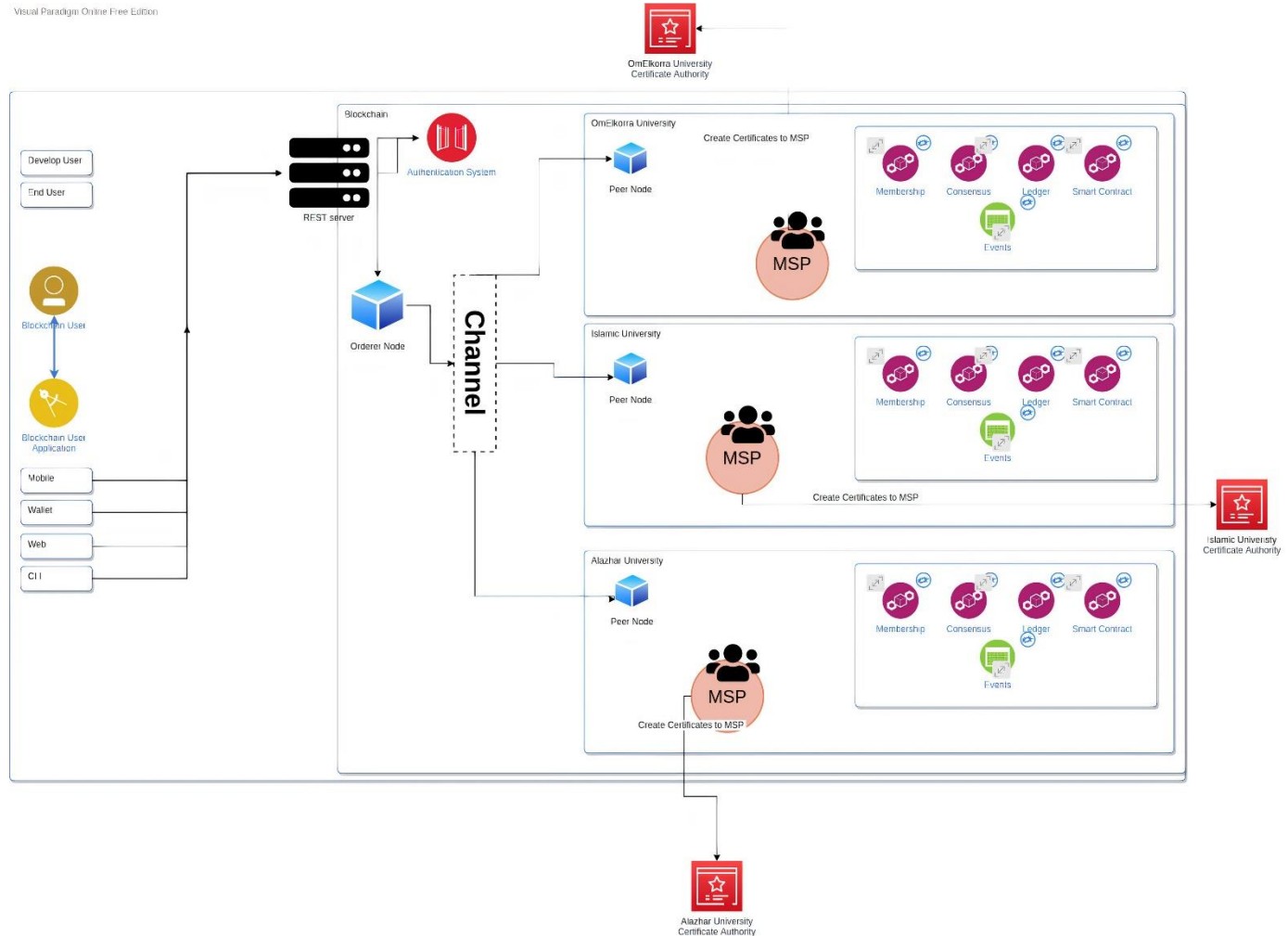

**Figure 1.** Solution Architecture includes three main entities connected through a communication channel controlled by order node.

## 4. Blockchain Implementation

We have a consortium (a federated hybrid wrap for the private blockchain where multiple organizations have the same authority over the blockchain) containing all non-orderers organizations. The implemented system has three organizations (peers). Each peer has policies (access) to read, write, and other administrative roles, such as adding new admins to the blockchain, backed by wallets for each peer to allow creating transactions.

This guide will refer to the proposed blockchain solution implemented in NodeJS as the server. The function is used to interact with the shared ledger as chaincode. The membership service provider (which defines the relationship between the identities of systems and enforces the predefined system policies) is an MSP. The system organizations as two different titles based on the current role; an organization user trying to add new/retrieve Hadiths as a client, an organization node which verifies transactions and hosts a copy of the distributed ledger as a peer.

For the main functionality in our proposed solution, which is the Hadith insertion, it goes as follows:

a.   Transaction Initiation

The client represented by the peers is sending a request to insert a Hadith, as in Figure 2. This request targets the system peers. Each peer should endorse the transaction as per the endorsement policy. Then the server creates a transaction proposal to call a

chaincode containing the Hadith parameters to update the ledger. Then the server signs the transaction proposal with the client's cryptographic key.

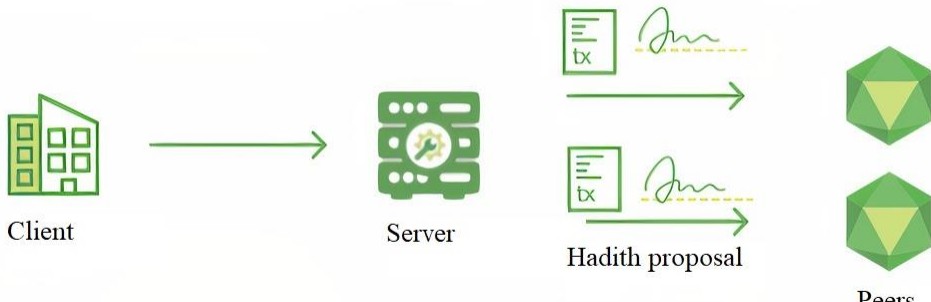

**Figure 2.** Hadith insertion request from the client.

b.    Peers signature verification and transaction execution

In Figure 3, the endorsing peers verify the novelty of the transaction proposal to prevent replay-attack and the authority of the client to perform this operation as per their policy using MSP. Then each endorsing peer starts a transaction execution simulation by executing a chaincode function using the proposal Hadith parameters against the current state of the ledger, which results in a response value, read, and write sets (which act as ledger state versioning for before and after execution). As a signed transaction proposal response, these values are signed with the peer keys and passed back to the server. At this point, the shared ledger is not updated.

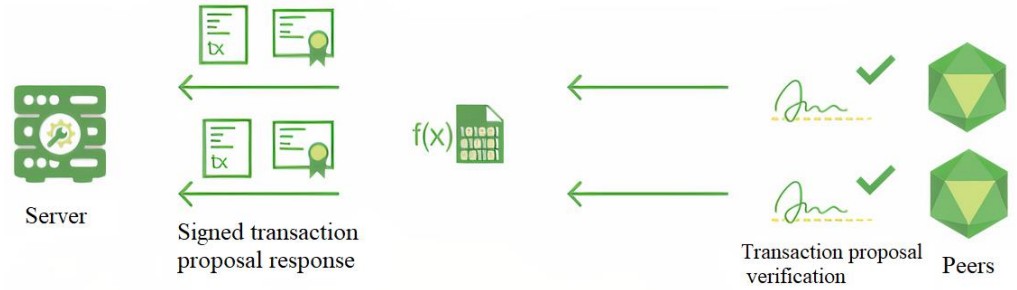

**Figure 3.** Transaction verification.

c.    Proposal responses inspection

In Figure 4, the server verifies each peer's signature and compares the proposal responses returned from each peer to ensure they match each other. Before the server starts submitting the transaction to the ordering service, it makes sure that the endorsement policy determined by the system architecture is fulfilled.

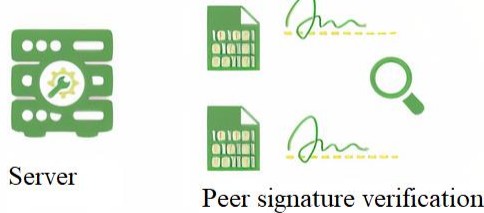

**Figure 4.** Endorsing peer signature verification.

d.    Transaction endorsement by the client

In Figure 5, the server broadcasts the transaction proposal to the ordering service through the communication system channel. The proposal contains the read/write sets,

the endorsing peers' signatures, and the channel id. The ordering service dumps the proposal content into newly created blocks of transactions.

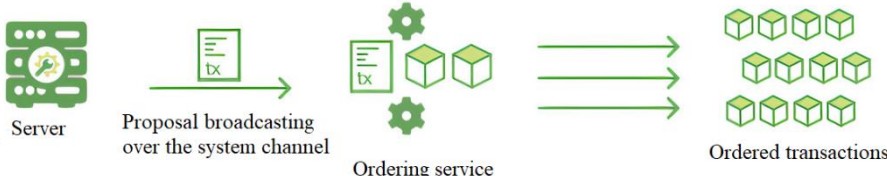

**Figure 5.** Transaction proposal broadcasting.

e.　Transaction validation and committing

In Figure 6, the ordering service ships the transaction blocks to all peers in the system. Once again, the peers ensure that the endorsement policy is fulfilled. The final check before updating the ledger is verifying that there are no changes to the ledger state embedded in the transaction block with the help of the read/write sets resulting from the transaction execution simulation at step (b). If no changes happen, the transactions are marked valid, else marked invalid.

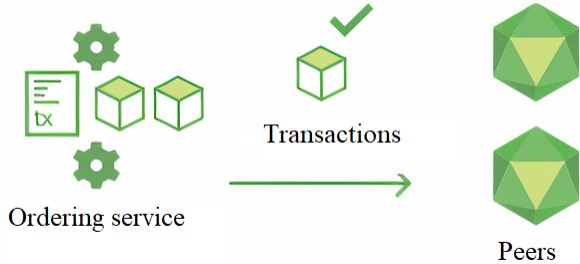

**Figure 6.** Transaction validation and committing.

f.　Database (Ledger) update

In Figure 7, each peer appends the block to the channel's chain, and each valid transaction containing the Hadith is committed to the current state database. Each peer emits an event to notify the client application that the transaction has been immutably appended to the chain. Another notification of whether the transaction was validated or not is published.

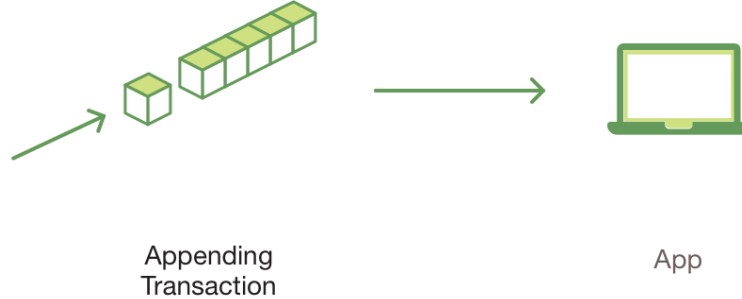

**Figure 7.** Ledger update.

## 5. Performance Evaluation

Our main aspects of the architecture evaluation are the performance evaluation and the data integrity guarantee. Performance evaluation is measured based on the following two factors: the time of reading from the blockchain and the time of writing new Hadiths to the blockchain. These are the two main operations in our proposed solution.

We used the official Apache benchmark tool supplied by httpd docker image. This tool sends requests to the blockchain server endpoints and measures response time, throughput, and other essential metrics. We focused on the response time precisely. In our experiments,

we represented each organization node with an official peer node docker image from the Hyperledger Fabric framework and the same for the orderer nodes. We are using NodeJS as a client-facing server for the REST operations. We used in our experiments a blockchain containing three peer nodes and one orderer node, all connected through one channel, with a block size of 190 KBs, which represents one Hadith sample in all experiments; each peer node represents an organization, and each organization has one user registered in the blockchain with administrative access. Using the default endorsement policy, we have a timeout for the block insertion equal to 2 s. We use the default endorsement policy, which accepts one valid signature from a verified peer node. Our primary evaluator was ten times the time taken to fulfill the number of requests made by the requester. Using the following two main parameters: the number of requests sent and the number of threads used to serve the requests, with multiple values representing our use cases, we evaluate the performance of the Hadith reading functionality in the blockchain.

The evaluation analysis was limited to using one machine to host the three nodes presented in the architecture. An official Hyperledger Fabric docker image represents each node corresponding to the node type. All nodes share the same machine specifications. Practically every node should be hosted on a different machine on a shared network.

The server specs are the following: Processor used is Intel® Core™ i5, 2.4 GHz, Dual-Core, and SSD with write throughput ~280 MBps and an 8 GB RAM device.

The performance evaluation for the blockchain solution was as follows:

1.  Read from the blockchain operation; we used multiple values for the number of requests, as mentioned in Table 2, varying from 100 requests up to 2000 requests. To benchmark the results from the read operation, we used a concurrency level of up to 2 threads (limited to the benchmarking machine specification), and we can see that the average response time, including accessing the ledger for each request, is relatively low when using the multi-threaded approach in Table 2.

**Table 2.** Average response time across single and double threads.

| Number of Requests | Average Response Time for 1 Thread (ms) (Mean) | Average Response Time for 2 Threads (ms) (Mean) |
| --- | --- | --- |
| 100 requests | 107.210 | 86.591 |
| 300 requests | 131.900 | 71.984 |
| 500 requests | 163.933 | 85.088 |
| 1000 requests | 171.249 | 91.522 |
| 2000 requests | 432.098 | 144.441 |

We can see the improvement in the read operation performance when using 2 concurrent threads for the 1000 requests. The significant improvement appears in the 2000 request experiment, where we can see how the concurrency led to a lower response time, opposite to the percentage of requests served.

In Table 3 we have more details about 2000 read requests where we calculate other values such as requests per second, which represents TPS (transactions per second) because 1 request equals 1 transaction, along with the minimum, maximum, median, and mean values in milliseconds for the response time, which are 71, 87, 79, and 194, respectively.

**Table 3.** Some metrics calculated from 2000 read requests.

| Metric | Value |
| --- | --- |
| Concurrency level | 2 |
| Time taken for tests | 17.376 s |
| Complete requests | 2000 |
| Requests per second | 11.51 [#/s] (mean) |
| Time per request | 86.881 [ms] (mean) |

2.  Write Hadiths to the blockchain operation; We used multiple values for the number of requests sent to the server and evaluated the time taken to serve these requests using

a single thread and two threads in parallel. The results in Table 4 show that using multithread decreases the time needed, increasing the overall blockchain performance as the Hyperledger Fabric supports concurrency by default.

**Table 4.** Average response time across single and double threads.

| Number of Requests | Average Response Time for 1 Thread (ms) (Mean) | Average Response Time for 2 Threads (ms) (Mean) |
|---|---|---|
| 100 requests | 3098.362 | 2652.926 |
| 200 requests | 2519.833 | 2269.546 |
| 300 requests | 2651.900 | 2531.984 |
| 500 requests | 2969.916 | 2345.715 |

Figure 8, how the performance is affected by the multithreading while inserting 100 requests, which leads to inserting 100 transactions into the blockchain, the same results for using 200 requests, and 500 requests.

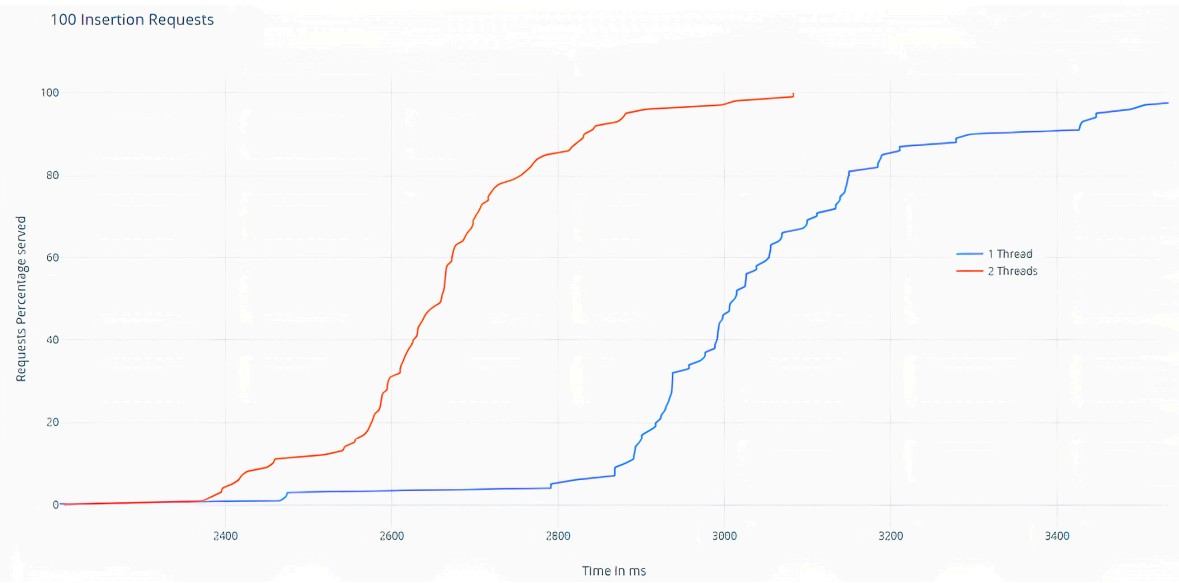

**Figure 8.** Requests served percentage for 100 write requests.

A logical bottleneck in the blockchain is the insertion, as the transactions need to be validated before insertion, and it takes time for all nodes to update their ledgers. The guarantee of data integrity is the main motive for using blockchain technology. In the centralized database architecture, the data are vulnerable to any manipulation if the credentials of any of the institutes involved in the system are exposed. An attacker can update the data in the database directly, and it will remain until this institute activity is identified as malicious activity and the credentials are revoked to prevent any further actions.

In the blockchain architecture suggested, we have two scenarios for any malicious activity when the credentials of any of the institutes involved in the system are exposed.

In Scenario-1, the institutes have already agreed on the following data transaction, and it has already been inserted into the decentralized database:

Any changes made to the data block will change the block hash, which is used as a block identifier and as each block has a reference to the previous block hash along with a timestamp [16]. All the other institutes in the network will detect the change, and to pass this change, the attacker needs approval from 50% + 1 of the network to update the decentralized database and change the chain of the blocks with the new entry, which ensures the immutability of the Hadiths.

Scenario-2: The institutes are in the process of voting on the following particular transaction: Our architecture is a permissioned blockchain where all participants are whitelisted according to a particular agreement (made offline between the institutes) and

since we expect most of the institutes to behave correctly, we use a more efficient consensus algorithm, which is the practical Byzantine Fault Tolerance (pBFT), [17], which works based on the concept of less than one-third of the network peers being faulty, represented as (f), so to tolerate this failure, the network peers should equal (3 ∗ f + 1) so (2 ∗ f + 1) peers can agree on the transaction reaching consensus. The BFT algorithms are proven to have better performance than the traditional blockchain consensus algorithms such as PoW (Proof of Work), while also highlighting scalability constraints [18–21].

### 6. Conclusions and Future Work

The evaluation of our proof-of-concept architecture implementation clarified both the pros and cons of using a blockchain-based solution for storing Hadiths. Even though the Hadith's insertion time is relatively longer than using a traditional centralized system, using a DLT (distributed ledger technology) such as Hyperledger Fabric prevents tampering and data manipulation. The system performs with acceptable latency in the read and write operations based on the application's usage. The application chosen for the experiments is read-driven and is expected to have few insertions after the seeding stage. Therefore, the solution proposed is using the main advantage of the distributed ledger and introducing, for the first time, the concept of using customized permissioned blockchain to preserve the historical texts. The solution also offers distributed management for the content inserted. The proposed implementation opens the door to more tuned solutions that enhance the performance results.

The future development includes extending the network to include sub-network blockchains for each organization. Each sub-network consists of trusted individual scholars. Each scholar contributes to the decision-making of the network to propose the transaction of each parent organization. As a potential improvement for the architecture, the consensus algorithm could be optimized to perform at scale, as the PBFT does not scale efficiently on a large number of nodes. Moreover, in future work, we can extend the Hadith blueprint to include more information, such as the different wordings for the same Hadith, all book references for this Hadith, different Isnads (Hadith attributions), and all different classifications for this Hadith. The system performance was limited to the testing machine specs. Therefore, we can then re-evaluate the blockchain based on the new size.

**Author Contributions:** Conceptualization, M.E., O.S., E.N., M.A. and M.T.; formal analysis, K.M.A., M.E.; investigation, O.S., E.N., M.A., M.E. and M.T.; methodology, M.E., O.S., E.N., M.A. and M.T.; project administration, M.E., M.A., O.S., E.N. and M.T.; software, K.M.A.; supervision, M.E., O.S., E.N., M.A. and M.T.; validation, K.M.A.; visualization, K.M.A.; writing—original draft, K.M.A.; writing—review and editing, K.M.A., M.E., M.A., O.S., E.N. and M.T. All authors have read and agreed to the published version of the manuscript.

**Funding:** The authors extend their appreciation to the Deputyship for Research and Innovation, Ministry of Education in Saudi Arabia for funding this research work through the project number 20/18.

**Institutional Review Board Statement:** Not applicable.

**Informed Consent Statement:** Not applicable.

**Data Availability Statement:** Data are available at https://github.com/khaledbnmohamed/hadith-blockchain (accessed on 12 February 2022).

**Conflicts of Interest:** The authors declare no conflict of interest.

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
