# Peer review of "A Secure Blockchain Framework for Storing Historical Text: A Case Study of the Holy Hadith"

_computers, doi:10.3390/computers11030042_

Round 1

Reviewer 1 Report

  1. The authors had used “Hadith” and “Haidth” as in Figure 2, so which one is correct? You need to guarantee the consistency.
  2. The related work should also mention the other applications of blockchain in a paragraph at the outset of the section to show that the authors have studied the literature and covered the entire area. There are many other applications of blockchain in the literature so, please have a better look at the literature and cover the area
  3. I’d like to see a table that compares the features of the blockchain in the literature and the proposed one. This can easily be fit in in Sec 2
  4. Where are the contributions of this paper? I cannot see them in Sec1, you need to add a list of contributions.
  5. Section 3 is short and needs more details about the proposed solution. Perhaps a formal model given that the visual is there.
  6. You also need to add more details about the evaluation environment and setup please.
  7. Proofread the work to fix any typos there. For example, in Page 3 line 111, what is this \cite{CoinMar- 111 ketCap}?

Author Response

We want to thank the reviewer for their thoughtful comments and their careful evaluation of the paper. We provide a point-to-point response to the reviewer's comments in the attached file. We hope that the revised version addresses the reviewer's concerns.

Reviewer 2 Report

The paper proposes a blockchain approach for securely archiving and validating the authenticity of historical texts. A permissioned blockchain is considered, in which only certain entities can approve the information.

Blockchain is a popular technology for which new areas of application are constantly emerging. In this context, the topic can be considered interesting and worth investigating. However, the actual approach presented in the paper cannot be considered very innovative. The authors are kindly asked to address the following issues:

  1. The presented approach is fairly standard in the blockchain context. There does not seem to be any novel contribution in the paper. In other words, what seems to differentiate the current paper from other papers is just the example that is given (namely of storing ancient texts). In this context the authors are kindly asked to highlight which are the aspects of the paper that they consider novel.
  2. The second section, which is dedicated to the literature review, also describes in the final part the proposed approach in almost half a page. The authors should consider shortening the description of their solution in this section and moving the remaining information (concerning the proposed approach) to the other sections of the paper. 
  3. The following paragraph appears twice in the paper: "In our proposed solution, the system used the prefabricated Hyperledger fabric’s tem-178 plate to create new networks. Starting with 3 organizations and one channel in Hy-179 perledger Fabric 1.4 as shown in Figure 1 and can be extended as the network needs".
  4. The following paragraph appears twice in the paper: "Each organization will apply its Hadith verification model for example using the power 168 of artificial intelligence and machine learning to build an Arabic natural language pro-169 cessing model based on the online Hadiths different datasets. This model can define the 170 degree of correctness of the Hadith and then consume the blockchain backend server 171 endpoints using the organization client to add the Hadith to the network. The Network is 172 built with the official Hyperledger Fabric images for peers, orderers, and chaincodes 173 where all peers have multiple policies (access) to different actions that allow writing and 174 reading from/to the blockchain. We have a consortium that contains all non-orderers or-175 ganizations and wallets for all peers in the network".
  5. The texts in Figure 1 are difficult to read. The authors are kindly asked to print the paper and check if they can read the text.

Author Response

(The authors gave the same response as above.)

Reviewer 3 Report

Dear Authors

the paper describes an interesting application of blockchain.

I'd suggest some amendements to improve the quality of your paper before publication:

  • add some more relevant keywords
  • section 2 "Related work" should be changed in "Literature background" and other similar applications, if any, of blockchain or other methods to preserve historical texts should be discussed in order to explain the originality/novelty of your application, as well as other applications of blockchain in other context
  • improve the quality of the figures
  • provide as per a table a summary of teh main pros and cons of your application
  • conclusions sections has to be improved, also by stressing the limitation of the study and proposing future developments
  • please revise the text for grammatical errors/typos

Good luck with your work

Author Response

(The authors gave the same response as above.)

Round 2

Reviewer 1 Report

The authors addressed the comments, thanks for that. I have no other comments to mention about this paper. I recommend accepting the paper in the current shape. 

Reviewer 2 Report

I would like to thank the authors for the changes made and for the comments. As highlighted in the previous review, from my point of view the originality of the paper is rather low.